# Strategies for achieving a healthy oral cholera vaccine market: Model-enabled scenario exploration of supply and demand dynamics

Donovan Guttieres⬚*, Carla Van Riet, Nico Vandaele, Catherine Decouttere

Access-To-Medicines Research Centre, KU Leuven, Leuven, Belgium

* donovan.guttieres@kuleuven.be

## Abstract

Following decades of progress, recent years have seen a resurgence of cholera. This has led to unprecedented demand for vaccines from the global emergency stockpile of oral cholera vaccines (OCVs), for outbreak and humanitarian use. As a consequence of chronic supply shortages, preventive vaccination has been suspended since 2022. Although strategic demand scenarios have been published for OCV, models that integrate OCV supply and demand across long time horizons are lacking. Therefore, a quantitative system dynamics model is presented to simulate OCV market dynamics between 2013–2035. The model considers the evolving OCV supply landscape as well as the impact of preventive efforts. Building on stakeholder-driven scenario design, simulations help identify leverage points to improve OCV market health and assess the individual and combined effect of interventions on accelerating cholera control. Specifically, country adoption of preventive vaccination programs and complementary investments in water and sanitation infrastructure are critical to reduce the risk of cholera. Although more resource-intensive, re-vaccination of at-risk populations helps sustain outbreak prevention. It also offers potential benefits such as increasing long-term demand predictability and the overall market size. These serve as important incentives to maintain supplier diversity, thus improving market health. However, since many cholera endemic countries rely on donor support to access OCV, budget constraints associated with reduced development aid can jeopardize programmatic ambitions. Interventions such as the use of rapid diagnostic tests and price competition of procurement can help meet country needs. Finally, market dynamics are influenced by the policies around when to resume preventive OCV use in endemic countries and 2-dose reactive vaccination. Specifically, inventory policies based on the current available stock level versus incoming stock in transit are compared. Aligning OCV supply with demand, both in time and quantity, will be critical to address immediate needs and support broader multi-sectoral activities towards cholera elimination.

**Data availability statement:** Data supporting results can be found in the supplemental file or through public online sources referenced in the paper. Two online databases are referenced: ICG cholera vaccine dashboard (https://www.who.int/groups/icg/cholera) and GTFCC OCV dashboard (https://apps.epicentre-msf.org/public/app/gtfcc).

**Funding:** DG reports financial support provided by Johnson & Johnson Innovative Medicine (https://innovativemedicine.jnj.com/us/global-public-health) via a donation to the Research Chair on Pandemic Preparedness at KU Leuven. The funder had no role in the study design, data collection and analysis, decision to publish, or preparation of the manuscript. All other authors did not receive funding for this work.

**Competing interests:** The authors have declared that no competing interests exist.

## Author summary

Cholera disease is caused by ingesting food or water contaminated by the *V. cholerae* bacteria. Multiple, converging factors have led to a resurgence of cholera, especially in cholera endemic countries across Sub-Saharan Africa and the Eastern Mediterranean region. However, cholera has also spread to previously unaffected countries, jeopardizing containment efforts and straining health systems. The growing burden of cholera has contributed to a growing mismatch between the supply and demand of oral cholera vaccines (OCV). This is seen by recurrent stockouts of the global emergency stockpile of OCV, primarily used for outbreak response, as well as constraints on preventive vaccination campaigns despite available programmatic funding and country interest. In this work, market health, the ability to align supply and demand both in quantity and time, is used as a proxy to assess progress on cholera control efforts and reduction in disease burden. A model is presented to capture OCV market dynamics between 2013–2035 to inform strategies aimed at improving OCV market health. Adopting preventive vaccination, with complementary multisectoral interventions, helps prevent outbreaks, improve predictability of OCV demand, and increase market attractiveness for suppliers. Market shaping strategies are critical to promote alignment of OCV supply with demand, as well as fully meet programmatic ambitions of countries.

## 1. Introduction

### 1.1. Resurgence of cholera

Multiple, converging factors have led to a resurgence of cholera: extreme weather events caused by climate change, humanitarian crises (e.g., forced displacement, armed conflicts, earthquakes), poorly planned urbanization, and inadequate access to basic infrastructure and services [1–3]. The situation is further exacerbated by chronic underfunding, limiting progress on cholera control efforts [4]. The number of countries reporting cholera cases has grown over time, from 35 in 2021 to 45 in 2023 [5]. From 2022 to 2023, there was a 71% increase in deaths and 13% increase in cases, largely in Africa, pointing to inadequate access to both prevention and treatment. Inadequate surveillance, non-specific clinical presentations, and reluctance by authorities to acknowledge cholera means the real burden is likely much higher [6]. In 2015, an estimated 1.3 billion people were at risk of cholera in endemic countries [7]. Like for other neglected diseases, the greatest burden is in countries with the weakest health systems and among the most vulnerable populations [8]. Socioeconomic losses associated with cholera can be severe, making vaccines especially cost-effective in regions with high incidence [9].

### 1.2. Role of oral cholera vaccines in global disease control

Oral cholera vaccines (OCV) play an important role in multisectoral interventions for the management of cholera. They target the O1 serogroup of the *V. cholerae* bacteria, which causes the majority of cases globally, helping protect populations

by reducing local transmission and contributing to herd immunity [10,11]. OCV use is recommended across two settings [12]: i) reactive use for outbreak response and humanitarian crises, and ii) preventive use for priority areas in endemic countries. A global OCV stockpile was established in 2013, with priority given to reactive use. The stockpile is continuously replenished to maintain a pre-defined target emergency reserve ($S^*$). Available doses beyond the $S^*$ can be used to meet demand in preventive settings. When supply is scarce, reliance on short-term, emergency vaccination delays long-term preventive efforts.

Vaccination with a safe and effective vaccine, although not a replacement for large-scale infrastructure investments in water, sanitation, and hygiene (WASH), bridges short-term outbreak response and long-term disease prevention. Therefore, the use of OCV helps achieve objectives outlined in a global strategy on cholera launched in 2017 by the Global Task Force on Cholera Control (GTFCC) [13]: 1) 90% reduction in outbreak deaths (relative to a 2017 baseline), and 2) elimination in up to 20 endemic countries by 2030. To achieve these objectives, all endemic countries are expected to identify Priority Areas for Multisectoral Interventions (PAMIs) and implement national cholera control plans (NCPs). Since 2023, Gavi, the Vaccine Alliance (GAVI) provides funding and technical assistance to implement multi-year phased preventive vaccination programs (pOCV), while continuing financial support for emergency deployment of doses from the global stockpile managed by the International Coordinating Group on Vaccine Provision (ICG) [14].

## 1.3. Key challenges

Achieving and sustaining a healthy market is not trivial, especially for vaccines with uncertain demand. The ability for OCV supply to meet demand has been increasingly difficult in recent years, with OCV market health categorized as 'unacceptable' based on GAVI's Healthy Markets Framework [15]. This can partly be explained by the following demand dynamics: i) growing evidence on the cost-effectiveness of OCV and feasibility of implementing large campaigns [16], ii) unanticipated surge in outbreaks leading to a sustained spike in requests for OCV use in reactive settings [17], and iii) growing interest in pOCV following expansion of GAVI funding. Additionally, numerous factors contribute to supply scarcity. The time needed for technology transfer, process scale-up, batch production, and lot release is long, leading to delays in closing the supply gap. Furthermore, since a majority of doses are delivered to reactive settings through UNICEF public procurement, low predictability of demand combined with thin profit margins have led to limited attractiveness for suppliers to enter and stay in the market [18]. As a result, there is an insufficient number and geographic diversity of suppliers (**Fig A in** S1 Appendix) despite sustained product innovation over the past few decades [19].

Current market dynamics accentuate challenges faced by decision-makers across the OCV value chain. The surge in outbreaks means doses are prioritized for reactive settings, with erratic orders and little visibility for suppliers to plan production in the long-term. Due to recurring shortages [20], preventive vaccination has been suspended since 2022, while WHO has recommended 1-dose reactive vaccination rather than the typical two-dose regimen [21]. Although dose sparing, single-dose vaccination has shown to confer reduced protective effect [22]. Furthermore, the current focus on emergency response makes it difficult for funders to anticipate future programmatic needs and for countries to implement NCPs, especially when faced with competing health priorities and limited resources. Recognizing these challenges, GAVI has published a roadmap to align mid- and long-term market-shaping objectives across partners and define necessary interventions to inform procurement strategies and decisions. Specifically, large demand is expected once pOCV resumes, estimated between 85–220 million doses annually on average over 10 years [15]. However, it is unclear how quickly new suppliers will enter the market and what impact procurement decisions will have on future market dynamics. In the meantime, the full programmatic potential of OCV remains out of reach, delaying cholera control efforts.

## 1.4. Research aims & contributions

There has been extensive modeling on the epidemiology of cholera, prediction of outbreaks, and impact of control measures, including OCV [23]. Models have been proposed to compare the impact of a one-dose versus two-dose OCV

vaccine regimen on disease transmission during an outbreak [24,25]. Given the importance of high vaccine coverage, one study explores the effect of population mobility and waning vaccine efficacy on vaccine-derived herd immunity [26]. Others look at the cost-effectiveness of different vaccination strategies, considering target age groups, geographic areas, and frequency of re-vaccination [27–29]. Most often, models focus on one (or few) geographic areas and a single outbreak event. To inform global OCV demand, some studies forecast the population at-risk of cholera across individual countries and regions [2,7,30]. No studies integrate OCV supply and demand at a global level over long periods.

This paper has several aims. First, compare the impact of different OCV supply and demand assumptions, including programmatic design, on market dynamics. Although this study focuses on OCV, it also considers synergistic actions (e.g., WASH investments) aimed at reducing the size of populations at risk of cholera. This is particularly relevant given the role multisectoral interventions have in modulating long-term OCV demand. Second, test the impact of market shaping strategies, defined by policies and interventions, on OCV market health. Here, market health is defined as the ability to align supply and demand both in quantity and time. This is critical to continuously meet programmatic needs and accelerate cholera control. Therefore, the study considers select key performance indicators (KPIs) such as doses requested and produced over time. Third, present a tool that can be used by practitioners to design, test, and compare scenarios, as well as researchers to determine avenues for future research.

These aims are accomplished by building a quantitative system dynamics (SD) model to simulate OCV market dynamics between 2013 and 2035. The model captures key interdependencies (e.g., feedback dynamics) between OCV supply and demand to better anticipate the individual and combined impact of interventions. It also considers the capacities and incentives of stakeholders in the OCV market. Additionally, a user interface allowed for rapid and interactive exploration, validation, and testing of scenarios together with stakeholders involved in market shaping activities. It displays key model and data assumptions, market shaping strategies, and KPIs of interest to stakeholders. By enhancing visibility of OCV supply and demand, the model promotes shared understanding of tradeoffs across strategies aimed at improving market health.

## 2. Methods

**Ethics statement**. This study has been reviewed and approved by the Social and Societal Ethics Committee (SMEC) of KU Leuven (G-2022–4999-R5). All insights from stakeholder interviews have been pseudoanonymized.

### 2.1. Study design & setting

SD modeling is particularly well-suited to provide strategic-level insights into complex problems characterized by non-linear behavior. It succeeds in capturing interactions between physical and institutional structures of a system, including relevant decision-making processes [31]. Since SD models capture the underlying structural drivers of observed behavior, they can effectively anticipate the long-term and unintended consequences of decisions, especially in complex settings [32]. The ability for SD to capture evolving feedback dynamics, often with long and variable delays, despite limited data made it more suitable than alternative modeling approaches such as econometric, agent-based, discrete-event, and optimization methods. Rather than testing interventions at a single point in time, SD models endogenously capture how policies alter system structures over time, often leading to unintended consequences and path-dependent outcomes. The complex interactions between multiple, concurrent interventions over time provide insight into system-level policy effects. The mechanisms inherent to SD models make this approach uniquely positioned to analyze future market dynamics and the dominant challenges addressed in this work, including understanding the tradeoffs between market shaping strategies under uncertainty. A participatory approach, through extensive stakeholder engagement, was used to design, develop, and validate the model. Model simulations are used to test the impact of deviations from baseline model assumptions and parameter values, based on user-defined scenarios.

The model captures historic (2013–2024) and future (2025–2035) dynamics of OCV supply and demand, providing visibility into both short- and medium-term market dynamics. This study considers the role of OCV, as well as complementary

multisectoral interventions, in accelerating objectives outlined in GTFCC's strategy towards global cholera control. Additionally, it aligns with the need for healthy markets outlined in the Immunization Agenda 2030 [33]. Specifically, the model scope relates to strategic priority 5 (outbreaks & emergencies) and 6 (supply & sustainability). It also aligns with the period and objectives defined in GAVI's Market Shaping Roadmap for OCV published in 2023 [15]. Finally, it responds to a recommendation by the Strategic Advisory Group of Experts (SAGE) on Immunization to model OCV supply scenarios to inform timing of pOCV [34].

Global OCV demand comes predominantly from cholera-endemic countries facing outbreaks and/or humanitarian crises. Most are low- and middle-income countries (LMICs), with vaccine procurement facilitated by UNICEF and funded by GAVI. Demand in reactive settings translates to country requests to use available doses from the revolving global OCV stockpile. These requests are reviewed and decided on by the ICG. In preventive settings, GTFCC's Country Support Platform (CSP) provides technical assistance for countries to apply for funding to support pOCV. These applications are reviewed by GAVI's Independent Review Committee (IRC). Once approved, decision letters (DLs) outline expected doses to be delivered to countries. UNICEF plays a critical role in operationalizing these DLs by procuring vaccines from qualified suppliers via purchase orders. GTFCC provides an allocation framework for fair distribution of doses destined for preventive use.

## 2.2. Data collection & stakeholder engagement

Data was collected from academic and grey literature, including public data sets. Additionally, semi-structured interviews with key stakeholders were conducted in English, lasting 1–2 hours, to inform this research. Interviewees were required to be actively involved in OCV supply and/or demand related activities, including research and funding, or engaged in processes that inform vaccine market shaping strategies. Through purposive and snowball sampling, interviewees were contacted between September 2024 and February 2025. In total, 36 interviews were conducted. Of these, 16 took place during early stages of model development, while 20 focused on model validation and to solicit feedback on the user interface. All had 10 + years of relevant work experience. Interview recordings and transcripts were securely stored and deleted post-transcription to ensure data privacy and confidentiality.

Stakeholder engagement throughout the study was critical to ensure model relevance and validity. Stakeholders contributed to the following: 1) understanding challenges related to OCV supply and demand; 2) identifying root causes of historic OCV shortages, including barriers and enablers to achieving healthy vaccine markets; 3) defining model boundaries for the research aims and ensuring inclusion of the most dominant feedback dynamics expected to drive future behavior; 4) clarifying publicly available data, estimating parameters, and validating model assumptions, structures, and behavior; 5) defining priority interventions and policies for scenario design, as well as KPIs to measure their relative performance; and 6) providing feedback on the model user interface. Wide representation of organization types (**Table A in S1 Appendix**) was important to learn from diverse perspectives and reduce bias.

## 2.3. Model structure & analysis

SD models are comprised of four interlinked building blocks: stocks, flows, auxiliary variables, and parameters/constants [35]. These building blocks are often organized into subsystems, representing different parts of a larger system. Connections are made between building blocks within and across subsystems, creating feedback loops that give rise to dynamics behavior. Arrays are used to efficiently replicate model structures across multiple elements within a dimension (e.g., product) and reduce visual complexity of the resulting model. During the simulation, at each time step (e.g., 1 day in this study), coupled differential equations are solved to approximate model values. Stella Architect (version 3.3, Isee Systems) was used to build the model and run simulations. Model output was processed to measure KPIs at each time step, annually, or at the end of the simulation.

Our SD model is comprised of several subsystems, each with a well-defined scope and relevant stakeholders (Table 1). The model reflects the most important dynamics in the OCV market, as found in literature and elicited through extensive

**Table 1. High-level overview of model subsystems.**

| Subsystem | Description and scope | Array dimension | Key stakeholders | Data sources |
|---|---|---|---|---|
| **Regulatory approval** *[exogenous]* | Time for suppliers to attain WHO pre-qualification. Consider 5 existing OCV products from 2013-2025 and 3 new OCV products by 2030 | Product | Suppliers, national regulatory authorities, WHO | [36] |
| **Global OCV policy** | Recommended use of WHO pre-qualified OCV products in both reactive and preventive settings; programmatic design of multi-year preventive programs in endemic countries (e.g., re-vaccination) | n/a | WHO, SAGE, ICG, GTFCC | Interviews, [12,37,38] |
| **Reactive OCV demand & orders** | Doses requested by countries and approved by ICG for use in reactive settings; translated to purchase orders to procure OCV from suppliers | n/a | Countries, ICG, GAVI | [39] |
| **Preventive OCV demand & orders** | Doses requested by countries and approved by GAVI IRC for use in preventive settings; translated to decision letters and then purchase orders to procure OCV from suppliers | Country | Countries, GTFCC, GAVI | [39] |
| **OCV product development** *[exogenous]* | Product characteristics that have implications on other model subsystems | Product | Suppliers, IVI, funders | [36] |
| **OCV production & supply** | Production process (drug substance, drug product, lot release) for OCV and supply to countries | Product | Suppliers, NRAs, UNICEF | Interviews, [38] |
| **Market attractiveness** | Dynamics of suppliers entering and leaving the market | Product | Suppliers | Interviews |
| **OCV order fulfillment** | Fulfilling approved OCV requests in both reactive and preventive settings to meet public health needs | n/a | UNICEF, Suppliers | Interviews, [40] |
| **Climate & conflict** *[exogenous]* | Aggregate impact of climate change and conflict, among other factors, on OCV demand | Country | n/a | [41] |

*Note:* Abbreviations – oral cholera vaccine (OCV), World Health Organization (WHO), Strategic Advisory Group of Experts on Immunization (SAGE), International Coordinating Group on Vaccine Provision (ICG), Global Task Force on Cholera Control (GTFCC), Gavi, the Vaccine Alliance (GAVI), International Vaccine Institute (IVI).

stakeholder discussions. The model considers reactive demand aggregated at a global level, while preventive demand is estimated for 47 endemic countries based on the time-dependent population at risk of cholera. Preventive demand also depends on the rate of pOCV adoption across countries and GAVI IRC approval. The model also captures the evolving supply landscape over time, defined as the number of suppliers and their corresponding production capacities. Once country requests are approved by the ICG or GAVI IRC, the decision to submit purchase orders to suppliers depends on policies that guide vaccine use across reactive and preventive settings. These policies are based on the available or future expected global OCV inventory. As a result, market dynamics that emerge provide insights on the extent to which supply aligns with demand. Additionally, three subsystems (labeled as 'exogenous') only include static parameters that inform dynamics captured in the model. A full description of model structures (stocks, flows, parameters), assumptions, and initial conditions can be found in S1 Appendix.

## 2.4. Structural & behavioral validity of model

Model validation is essential to build confidence in simulation results. This study's focus on informing OCV market shaping strategies means validation centered around the directionality and relative magnitude of change under different settings, rather than specific predictions or optimization. Since the goal of SD models is to explain behavior through causal structures, validation standards are typically more rigorous than many other methodologies [42]. Unit consistency of the model

was checked. When published data was limited, parameter values and ranges, relationships, and behavior were validated by stakeholders with relevant domain knowledge. Since future OCV market dynamics are expected to be significantly different from past behavior, it was particularly important to validate model behavior under likely supply and demand scenarios. Additionally, the large-scale preventive use of OCV is new for many endemic countries. Therefore, learnings were taken from vaccine markets similar to OCV, including those for yellow fever and meningococcal vaccines. Model improvements were made iteratively following validation sessions with stakeholders.

Multiple structure and behavior tests were conducted, starting with each subsystem individually and repeated for the full model [43,44]. This involved sensitivity analysis of key parameters, testing extreme conditions to capture dynamics not seen in the baseline scenario, and comparing simulated and observed behavior. Behavior reproduction was done for selected KPIs such as doses requested and produced, including testing the individual and combined effect of feedback loops. Additionally, weekly reported data on the global OCV stockpile level was compared to simulation results between 2021 and 2024. Boundary adequacy tests, largely through stakeholder interviews, helped ensure that the level of aggregation is appropriate for the research aims defined and that dynamics most responsible for driving behavior are captured in model structures. Taken together, these validation tests ensure the model matches observed behavior for the right reasons, increasing interpretability of results.

### 2.5. Interactive learning environment

In this study, a computer-based user interface was developed to provide an opportunity for key stakeholders to define, test, and compare model scenarios. The interactive learning environment (ILE) was developed with Stella Architect, allowing easy interaction with and adaptation of variables in the model. First, stakeholder interviews helped define user needs, requirements, and preferences. Second, a mockup of the ILE was designed using sketches and wireframes to ensure a clear user experience. Third, various prototypes were made with increasing functionality and unique user flows. Input (e.g., knobs, sliders, switches) and output (e.g., graphs, numeric display) options were explored. Fourth, user testing was done with stakeholders to solicit feedback on the ILE, especially around variables to be defined by users and KPIs to be displayed. Fifth, the ILE was further refined. Finally, a user guide, including a test case, was developed to facilitate independent use. Considering evolving OCV dynamics, the ILE was built in a modular way to rapidly adapt to new user requirements and realities.

## 3. Stakeholder-driven scenario design

Given the broad scope of the model, a large number of scenarios can be simulated. Scenarios were selected and designed to focus on real-world relevance, with assumptions based on extensive stakeholder discussions and available data. Scenarios also capture sensitivity analysis of KPIs to changes in critical, yet uncertain, parameters. Depending on the scenario, one or multiple parameters are changed simultaneously. Scenarios presented are of increasing complexity, gradually considering additional interventions and policies. First, scenarios test the impact of different supply, demand, and programmatic assumptions on OCV market dynamics. Then, two sets of market shaping strategies are explored: interventions to remedy programmatic gaps when assuming budget constraints and policies for resuming full OCV use in both reactive and preventive settings. Although parameters are estimated based on available knowledge, different assumptions could be made to reflect heterogeneity across countries and products. Parameters not explicitly mentioned are assumed to remain the same across all scenarios and can be found in S1 Appendix.

### 3.1. Supply and demand assumptions

Three possible future supply settings are defined (Table 2). For each scenario, only one supply setting is selected. We assume each supplier only produces one OCV product. In the baseline supply setting (BSS), of the five OCV products

**Table 2. Definition of potential future supply settings.**

| | | Euvichol-S | Product 1 | Product 2 | Product 3 |
|---|---|---|---|---|---|
| **Baseline Supply** (BSS) | [1] $T_{PQ}$ | April 2024 | | | |
| | [2] $C_{max}$ | 75 M | | | |
| | [3] $M_{min}$ | n/a | | | |
| | [4] $I_{max}$ | n/a | | | |
| **Optimistic Supply** (OSS) | $T_{PQ}$ | April 2024 | July 2025 | July 2027 | July 2028 |
| | $C_{max}$ | 75 M | 75 M | 40 M | 15 M |
| | $M_{min}$ | 5 M | 20 M | 10 M | 1 M |
| | $I_{max}$ | 100 M | 100 M | 50 M | n/a |
| **Pessimistic Supply** (PSS) | $T_{PQ}$ | April 2024 | January 2026 | January 2028 | January 2030 |
| | $C_{max}$ | 75 M | 15 M | 40 M | 75 M |
| | $M_{min}$ | 5 M | 1 M | 10 M | 20 M |
| | $I_{max}$ | 100 M | n/a | 50 M | 100 M |

All potential future products are assumed to have 2 strains (*V. cholerae* O1 Inaba and O1 Ogawa), similar to the simplified formulation found in the currently produced Euvichol-S.

*Note:* [1]$T_{PQ}$: Time when a supplier expects to receive WHO pre-qualification for its OCV product and becomes eligible for public UNICEF tenders; [2]$C_{max}$: Maximum annual production capacity; [3]$M_{min}$: Minimum annual market size for each supplier to stay in the market, adjusted by the number of suppliers in the market; [4]$I_{max}$: Maximum excess inventory across all suppliers for each supplier to stay in the market; M: millions of doses. See S1 **Appendix** for more details on assumptions.

with WHO PQ (Dukoral, Shanchol, Euvichol, Euvichol-Plus, Euvichol-S), we assume only Euvichol-S is produced as of 2025. The incumbent supplier (EuBiologics, South Korea) stays in the market throughout the entire simulation period (2025–2035), with a maximum capacity of 75 million doses per year as of 2025 [45]. For the two other supply settings, the model is supplier agnostic, aiming to capture diverse supplier profiles.

In the optimistic supply setting (OSS), three new suppliers enter the market by 2028. To account for gradual capacity phase-in, the maximum production capacity (measured in OCV doses) for product 1, 2, and 3 is reached by 2028, 2030, and 2031, respectively. We model product 1 supplier as the most risk-averse and likely to exit the market, while product 3 supplier is least likely. Market attractiveness is determined based on annual review of the OCV market size and continuous review of excess inventory across all suppliers. In the pessimistic supply setting (PSS), three new suppliers enter the market by 2030, reaching their maximum capacity after three years. Although long-term production capacity is the same for OSS and PSS, they differ in their ability to rapidly remedy supply gaps, especially in anticipation of pOCV adoption. Assumptions on the number of new suppliers and their relative settings is based on publicly reported OCV development and technology transfer efforts. Capacity phase-in reflects historically observed behavior. Since OCV is a high-priority vaccine for WHO PQ [36], the majority of the reported scenarios consider the OSS setting.

The first four scenarios focus on different OCV supply and demand assumptions (**Table 3**). The 'baseline' scenario (scenario 1) assumes minimal intervention and a BSS. Vaccination is only considered in reactive settings, with an annual increase in OCV demand driven entirely by risk factors (e.g., climate) for cholera outbreaks and humanitarian emergencies. We assume the effect of climate and conflict to be the same in all scenarios. Scenarios 1–3 consider slow market entry of suppliers (PSS). The 'fixes that fail' scenario (scenario 2) assumes one-time pOCV with no re-vaccination, in the absence of multisectoral interventions. Adoption of pOCV in endemic countries is considered slow: 1–3 countries apply for funding each year between 2024–2033. For each scenario, the exact timing is based on a probability distribution. In the 'multisectoral approach' scenario (scenario 3), slow pOCV adoption is simulated with re-vaccination every 3 years. Also, WASH investments lead to a 50% reduction in the at-risk population between 2028 and 2035. These expected delays reflect challenges around capital-intensive investment requirements and need for regular maintenance. [46]. Scenario

**Table 3. Key parameter assumptions for OCV supply and demand in scenarios 1-9.**

| | Scenario description | Supply setting *See Table 2* | Reactive demand setting | | | Preventive demand setting | | | | | |
|---|---|---|---|---|---|---|---|---|---|---|---|
| | | | [a] $E_{cco}$ | [b] $E_{pocv}$ | [c] $E_{hss}$ | [d] $E_{ccp}$ | [e] $E_{pop}$ | [f] $N_p$ | [g] $A_{pocv}$ | [h] $P_{rev}$ | [i] $E_{wash}$ |
| 1 | Baseline | BSS | ✓ (3%) | | | ✓ | ✓ | | | | |
| 2 | Fixes that fail | PSS | ✓ (3%) | ✓ | | ✓ | ✓ | ✓ (3) | slow | | |
| 3 | Multisectoral approach | PSS | ✓ (3%) | ✓ | ✓ | ✓ | ✓ | ✓ (3) | slow | ✓ | ✓ (50%) |
| 4 | Accelerating investments | OSS | ✓ (3%) | ✓ | ✓ | ✓ | ✓ | ✓ (3) | slow | ✓ | ✓ (80%) |
| 5 | 2-phase, slow adoption | OSS | ✓ (3%) | ✓ | ✓ | ✓ | ✓ | ✓ (2) | slow | ✓ | ✓ (80%) |
| 6 | 4-phase, slow adoption | OSS | ✓ (3%) | ✓ | ✓ | ✓ | ✓ | ✓ (4) | slow | ✓ | ✓ (80%) |
| 7 | 2-phase, fast adoption | OSS | ✓ (3%) | ✓ | ✓ | ✓ | ✓ | ✓ (2) | fast | ✓ | ✓ (80%) |
| 8 | 3-phase, fast adoption | OSS | ✓ (3%) | ✓ | ✓ | ✓ | ✓ | ✓ (3) | fast | ✓ | ✓ (80%) |
| 9 | 4-phase, fast adoption | OSS | ✓ (3%) | ✓ | ✓ | ✓ | ✓ | ✓ (4) | fast | ✓ | ✓ (80%) |

*Note:* [a]$E_{cco}$: Annual change in the volume of requests for OCV use in reactive settings due to climate change and conflicts (exogenous). [b]$E_{pocv}$: Annual change in the volume of requests for OCV use in reactive settings due to preventive vaccination, assuming a 2-dose regimen and 50% target coverage of the at-risk population. [c]$E_{hss}$: Annual change in the volume of requests for OCV use in reactive settings due to health system strengthening and outbreak preparedness (exogenous), assuming a sigmoid function, [d]$E_{ccp}$: Annual change in each country's population at risk of cholera due to climate change and conflicts, based on Notre Dame Global Adaptation Initiative's vulnerability index (exogenous); [e]$E_{pop}$: Annual change in each country's population at risk of cholera due to demographic changes, accounting for estimates for access to improved sanitation (exogenous); [f]$N_p$: Number of phases in multi-year preventive vaccination programs (exogenous); [g]$A_{pocv}$: Adoption rate – slow or fast – of preventive vaccination programs among endemic countries (exogenous); [h]$P_{rev}$: Policy to pursue re-vaccination after countries complete an initial multi-year phased preventive program (exogenous); [i]$E_{WASH}$: Overall change in each country's population at risk of cholera between 2028–2035 (exogenous), assuming an exponential decay function. See S1 Appendix for more details on assumptions.

3 considers the impact of health system strengthening and outbreak preparedness. Comparatively, the 'accelerating investments' scenario (scenario 4) assumes greater reduction (80%) of the at-risk population due to improved WASH and a more ambitious timeline for suppliers (OSS). It is defined as the 'preferred' demand scenario to accelerate progress towards GTFCC objectives.

Building on assumptions in the preferred demand scenario, scenarios 5–9 further explore the extent to which two specific strategies, i.e., adapting the rate of pOCV adoption and vaccine roll-out (phases), influence market dynamics. The number of phases in a given preventive OCV program determines the size of each campaign until the target population is fully vaccinated. Scenarios 5 and 6 test the impact of shifting to a 2-phase or 4-phase program, respectively, compared to the 3-phase assumption in scenarios 1–4. Scenarios 7–9, investigate the impact of fast pOCV adoption: 3–5 countries apply for funding each year between 2024–2030. Several assumptions are made across all scenarios. In line with current practice, there are 12 months between each pOCV phase. Additionally, only GAVI-eligible countries classified as high- and medium-likelihood adopters, based on GAVI's strategic demand scenarios for OCV, are considered for preventive vaccination. Starting in 2025, we assume pOCV resumes if the available inventory across all suppliers is at least 15 million doses. No constraint is placed on the number of shipments required to fulfill an order. The current dose sparing policy (1-dose vaccination) remains in effect in reactive settings, while S* is maintained at 5 million doses.

### 3.2. Market shaping strategies under budget constraints

In previous scenarios, costs associated with vaccine procurement and implementation support are calculated, with no budget constraints. However, economic challenges and austerity measures across countries have led many GAVI donors to rethink funding commitments [47,48]. Therefore, several scenarios are defined to capture the impact of key

feedback mechanisms and strategies on market health (**Table 4**). Scenarios 10–20 have the same supply, demand, and programmatic settings as scenario 4. However, they assume a total available programmatic budget of USD 2.75 billion between 2013 and 2035, compared to no budget constraint in scenarios 1–9. As a result, approved DLs cannot be fulfilled once the budget threshold is passed. Given this budget constraint, the model considers the individual effect of 1) adopting rapid diagnostics tests (RDT) (scenario 11), 2) price competition in procurement (PCP) as new suppliers enter the market (scenario 12), and 3) ordering behavior (OB) of countries (scenario 13) on the number of doses requested. Scenario 14 looks at the combined effects of all three.

Since 2023, GAVI provides support to improve diagnostic testing capacity for cholera [49]. Scenarios with RDT assume 16 countries have adopted rapid diagnostic testing by 2025, reducing pOCV demand by 25% in each. Price competition emerges as new suppliers enter the market, leading to a lower average awarded price per dose during UNICEF procurement, assumed to be USD 1.35 per dose as of 2025. We assume each new supplier leads to a 10% decrease in the awarded price, with a minimum of USD 1.00 per dose. Finally, historical data shows an effect of supply levels on ordering behavior [50]. The model considers this effect separately for reactive and preventive settings. In reactive settings, when the inventory level is below $S^*$, the number of ICG-approved doses is on average 35% lower than the volume requested without supply constraints. In preventive settings, the model assumes that the accumulation of pending DLs due to insufficient supply leads countries to further prioritize target populations for pOCV. They are unlikely to entirely cancel requests given investments made to develop NCPs and identify PAMIs.

As the supply situation improves, a recurring question highlighted by stakeholders is how to design policies to resume both pOCV and 2-dose reactive vaccination (BOOST), as well as anticipate tradeoffs between decisions. Therefore, scenarios 15–20 test 6 inventory policies, while accounting for the combined effect of budget constraints, RDT, PCP, and OB

Table 4. Definition of interventions under budget constraints, including the use of rapid diagnostic tests, price competition during procurement, and ordering behavior.

| | | Feedback dynamics | | | | Policy for resuming preventive vaccination (pOCV) | | Policy for resuming 2-dose reactive Vaccination (BOOST) | |
|---|---|---|---|---|---|---|---|---|---|
| | Scenario description | [a] BC | [b] RDT | [c] PCP | [d] OB | [e] Criterion | [f] Threshold (million) | Criterion | Threshold (million) |
| 10 | Budget constraint | ✓ | | | | ✓ $P_i$ | 15 | n/a | |
| 11 | Budget constraint+rapid diagnostics | ✓ | ✓ | | | ✓ $P_i$ | 15 | | |
| 12 | Budget constraint+price competition | ✓ | | ✓ | | ✓ $P_i$ | 15 | | |
| 13 | Budget constraint+ordering behavior | ✓ | | | ✓ | ✓ $P_i$ | 15 | | |
| 14 | Budget constraint+RDT+PCP+OB | ✓ | ✓ | ✓ | ✓ | ✓ $P_i$ | 15 | | |
| 15 | Full OCV use – policy 1 | ✓ | ✓ | ✓ | ✓ | ✓ $P_i$ | 15 | ✓ $P_i$ | 5 |
| 16 | Full OCV use – policy 2 | ✓ | ✓ | ✓ | ✓ | ✓ $P_i$ | 15 | ✓ $P_i$ | 10 |
| 17 | Full OCV use – policy 3 | ✓ | ✓ | ✓ | ✓ | ✓ $P_{dpwip}$ | 30 | ✓ $P_i$ | 5 |
| 18 | Full OCV use – policy 4 | ✓ | ✓ | ✓ | ✓ | ✓ $P_{dpwip}$ | 30 | ✓ $P_i$ | 10 |
| 19 | Full OCV use – policy 5 | ✓ | ✓ | ✓ | ✓ | ✓ $P_{totalwip}$ | 60 | ✓ $P_i$ | 5 |
| 20 | Full OCV use – policy 6 | ✓ | ✓ | ✓ | ✓ | ✓ $P_{totalwip}$ | 60 | ✓ $P_i$ | 10 |

Given these combined effects, inventory policies for resuming preventive vaccination and 2-dose reactive vaccination are defined based on three different decision criteria. All other parameters are the same as in scenario 4.

*Note:* [a]**BC**: Effect of budget constraints on fulfilling decision letters and OCV delivery following approved requests; [b]**RDT**: Effect rapid diagnostic testing on the size of target populations for pOCV in endemic countries; [c]**PCP**: Effect of price competition, based on the number of suppliers in the market, on UNICEF's awarded price per dose per year; [d]**OB**: Effect of supply levels on ordering behavior in both reactive and preventive settings; [e]**Criteria**: Policies for resuming preventive vaccination and 2-dose vaccination in reactive settings, based on actual inventory levels in stock ($P_i$), batches to be released within the next ~6 weeks ($P_{dpwip}$), or batches to be released within the next ~10 weeks ($P_{totalwip}$); [f]**Threshold**: Minimum value, measured in millions of doses, for the criteria to be met, leading to a policy change (e.g., resuming pOCV). See S1 **Appendix** for more details on assumptions.

(as in scenario 14). Three decision criteria are considered for resuming pOCV: available inventory in stock, batches to be released within ~6 weeks (DP WIP), and batches to be released within ~10 weeks (total WIP). The WIP includes available inventory in stock. For BOOST, the inventory level in stock is the only criterion used. A policy is defined by combining one decision criterion with a minimum threshold to be met or exceed.

## 4. Results

### 4.1. OCV market dynamics

Key dynamics and processes observed in the OCV market are captured in a high-level conceptual framework shown in **Fig 1**. It captures interactions between model subsystems defined in Table 1, including demand in both reactive and preventive settings, as well as OCV supply and order fulfillment. One example is the link between order fulfillment and market attractiveness. Suppliers are likely to stay in the market when the rate of order fulfillment is high, unless future

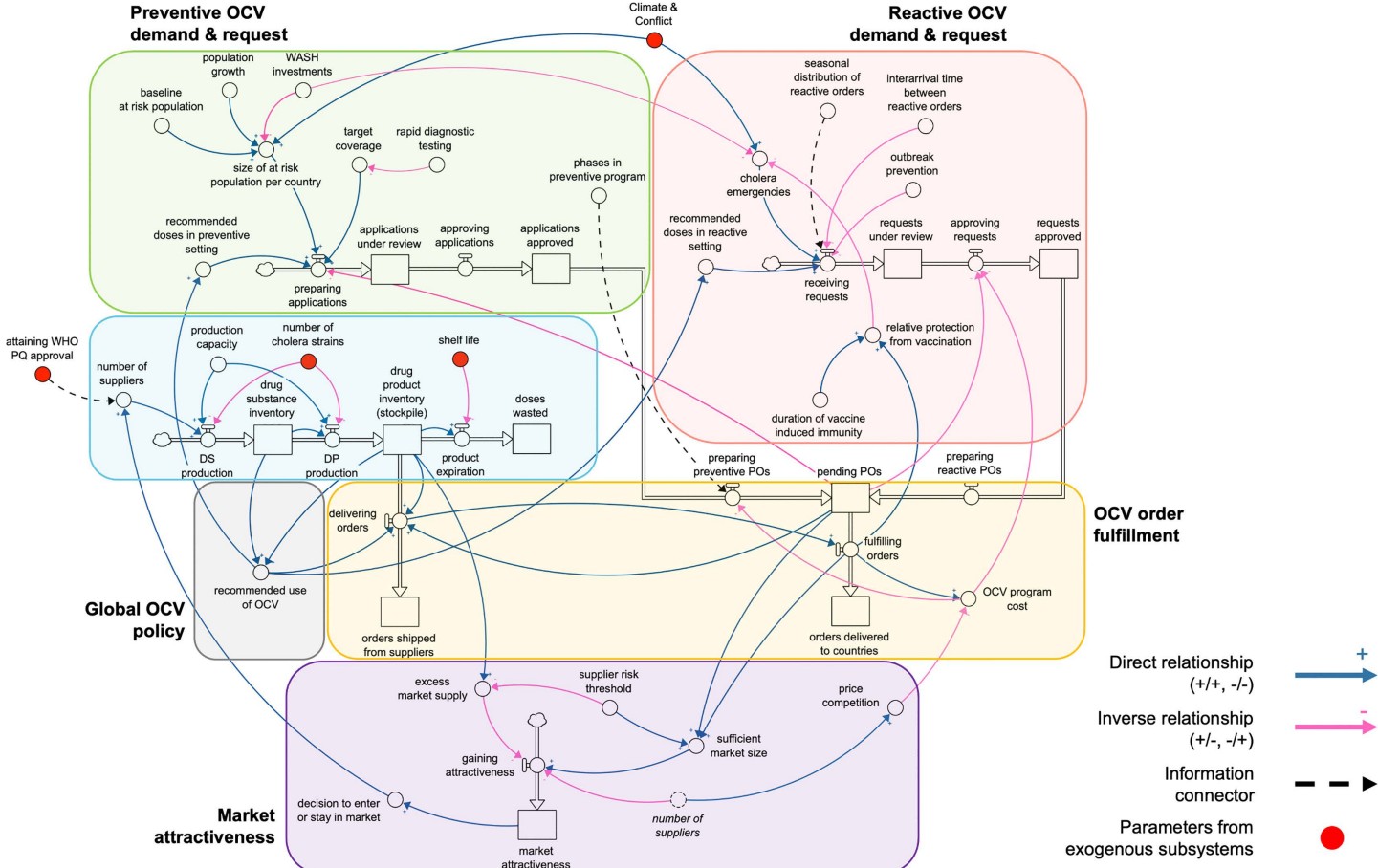

**Fig 1. Conceptual stock-and-flow model of key subsystems, interactions between them, and feedback dynamics in the model.** Variables from exogenous subsystems are shown in red. Stocks (boxes) capture quantities (e.g., doses) in the system over time, based on the rate of inflow and outflows. Interactions between different system elements are captured by arrows with polarities that refer to a direct or inverse relationship. Non-linear behavior emerges from delayed effects between variables, as well as feedback loops – when an initial change in a variable causes a chain reaction that affects the starting variable.

expected demand is low or the market is saturated. This complex behavior is captured through multiple, endogenous relationships that evolve over time. Moreover, the model allows decision-makers to identify the root cause of dynamics, for example if a supplier leaves the market due to insufficient demand or excess inventory. The impact of supply levels on global OCV policies is another important dynamic captured. The decision to resume pOCV or BOOST has direct impact on doses requested and delivered. It also influences predictability and materialization of future orders and overall market size. A shift towards preventive vaccination attenuates demand in reactive settings and allows suppliers to better anticipate capacity requirements for pOCV. Some relationships are captured exogenously, including the impact of climate and conflict on OCV demand and that of complementary multisectoral interventions such as WASH and health system strengthening.

Several KPIs (**Table B in** S1 Appendix) are defined to analyze simulated (quantified) system behavior over time, under different supply and demand settings. KPIs focus on OCV market health and the ability for vaccines to accelerate cholera control efforts (**Fig 2**).

## 4.2. Model validation

Stakeholders expressed that the model conceptually and structurally captures the main issues currently seen in the OCV market. Furthermore, the model replicates historical UNICEF-reported inventory levels of the global stockpile (**Fig B in** S1 Appendix), which emerges from complex supply and demand dynamics. Statistical measures such as the root mean squared error, $R^2$, and Theil inequality coefficient are reported in S1 **Appendix**. As the OCV market evolves, model adaptations can be made to reflect learnings.

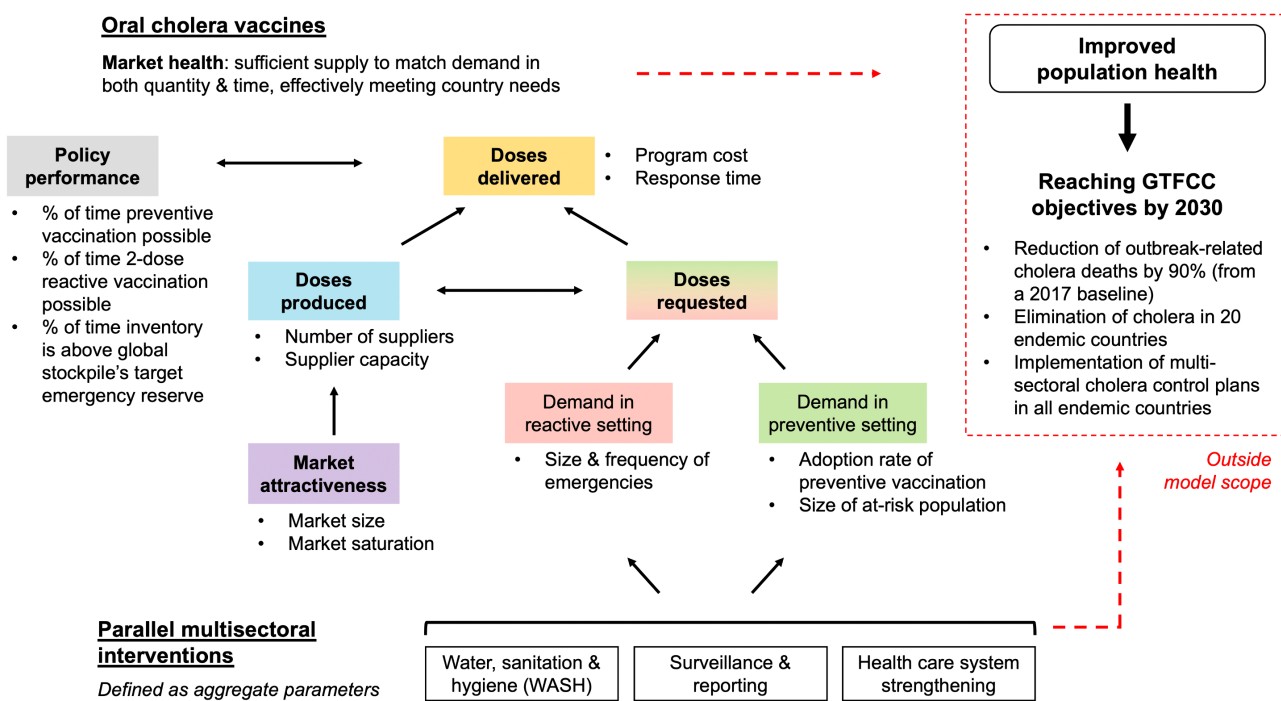

**Fig 2. Overview of key performance indicators across model subsystems and their relationship with overall objectives defined by the Global Taskforce on Cholera Control (GTFCC).**

## 4.3. Scenario analysis: tradeoffs across OCV supply and demand assumptions

Between 2013 and 2024, 158 million and 47.3 million doses were delivered to countries for reactive and preventive use, respectively [37]. Fig 3 presents OCV supply and demand, as well as the number of qualified OCV suppliers in the market, for scenarios 2–4. In the baseline (scenario 1), OCV demand is solely driven by outbreaks and humanitarian crises, leading to a total of 881 million doses requested for use in reactive settings between 2013 and 2035. Annual production of 75 million doses in the BSS is sufficient to fulfill demand, but the limited predictability of demand means suppliers continue producing at risk. Stakeholders point to the lack of visibility on future demand as the biggest failure of the OCV market. Continued dependence on outbreak response will likely translate to recurring outbreaks and associated socioeconomic disruptions. Re-vaccination combined with multisectoral interventions leads to a significant reduction in the cumulative number of doses requested in reactive settings relative to one-time pOCV. Specifically, a 27% and 30% reduction is seen

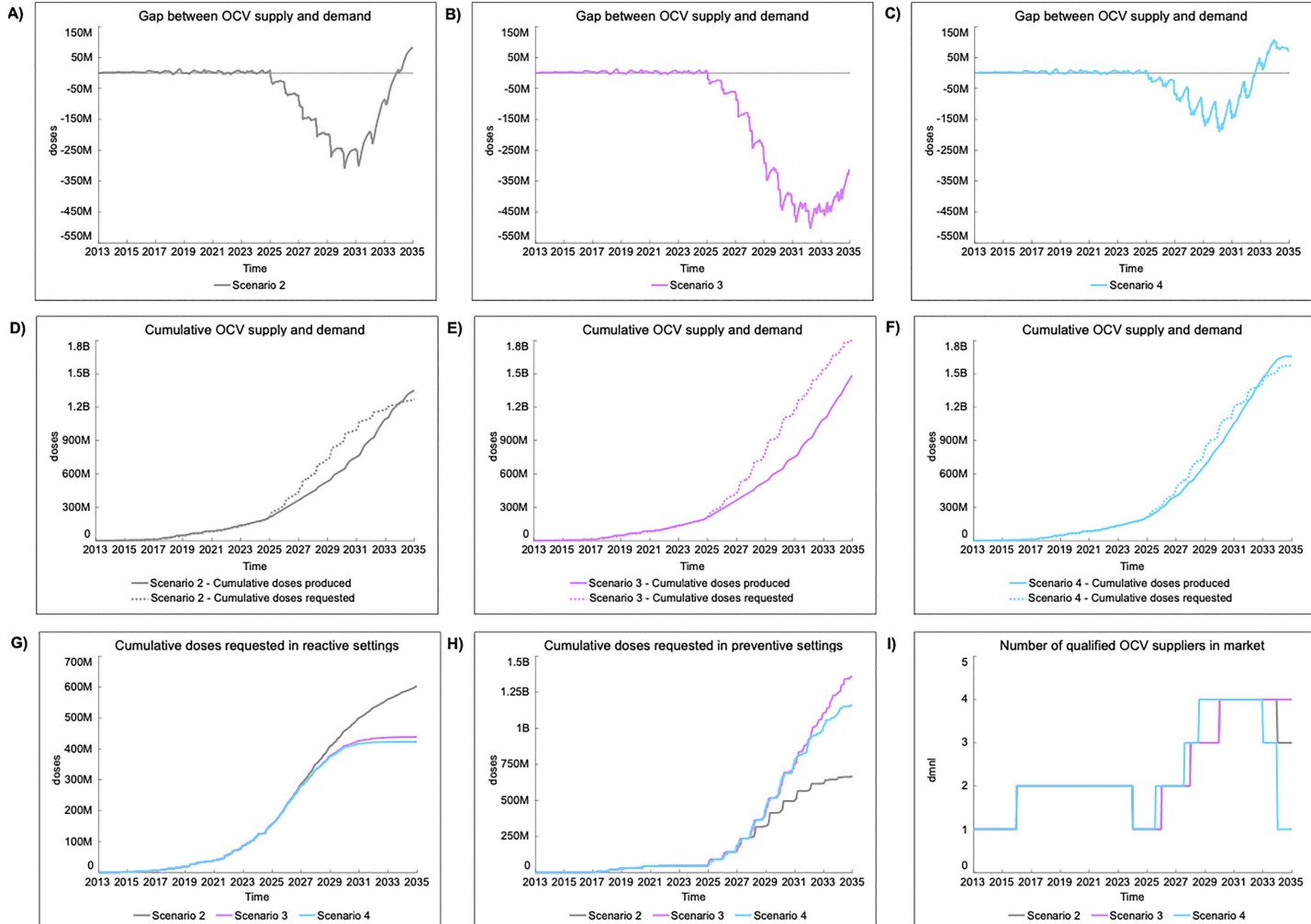

**Fig 3. Key performance indicators for scenario 2 (grey), 3 (purple), and 4 (blue).** A-C) Gap between OCV supply and demand over time; D-F) Cumulative OCV supply (solid line) and demand (dotted line); G) Cumulative doses requested in reactive settings; H) Cumulative doses requested in preventive settings; **I)** Number of OCV suppliers with WHO pre-qualification supplying to the global stockpile through UNICEF public procurement. Note: The range for X-axes is the same for all figures (2013-2035), while the range for Y-axes depends on the KPIs displayed.

in scenario 3 and 4, respectively, relative to scenario 2. However, the cumulative number of doses requested in preventive settings is significantly higher with re-vaccination. These dynamics present important tradeoffs. First, demand predictability increases when pOCV is considered, with at least 50% of annual doses delivered reaching preventive settings by 2030 in scenario 3 and 2028 in scenario 4 (**Fig C in** S1 Appendix). Second, to effectively reduce the supply gap throughout the simulation period, new suppliers need to enter the market earlier and scale-up capacity faster. For scenario 4, reaching an annual supply capacity of ~200 million doses by 2031 helps close supply gaps in 2032. However, as supply accumulates and the annual OCV market size declines (Fig D in S1 Appendix), some suppliers will leave the market prior to 2035 due to reduced market attractiveness. Third, although scenario 4 is preferred to maximize timely availability and programmatic use of OCV, it comes at a higher cost (Fig E in S1 Appendix). The estimated ~USD 3.3 billion needed for OCV procurement and implementation support throughout the simulation period is 19% and 6% higher than expected costs for scenario 2 and 3, respectively. Since public procurement with GAVI funding depends on donors with competing priorities over time, understanding the distribution of costs across GAVI's 5-year funding cycles is important. For scenario 2 and 3, the highest cost period is between 2031–2035 compared to 2026–2030 for scenario 4.

**4.3.1. Implications of programmatic design.** Multi-year preventive vaccination programs, especially for endemic countries with large target populations, lead to spikes in demand. Therefore, programmatic design can play an important role in smoothing demand, especially when switching from 2 to 4 phases (**Fig 4**). In a supply-scarce environment, with a long wait time until new suppliers enter the market, extending pOCV over more years reduces delays between country requests and order fulfillment by suppliers. Better aligning demand with future expected supply also leads to improved management of the stockpile, as seen by longer periods when inventory is above $S^*$ (**Table C in** **S1 Appendix**). However, unintended consequences can emerge from prolonged pOCV, as seen in scenario 6 and 9. When pOCV is stretched over too many years, countries are less likely to initiate second and third round vaccination in PAMIs. In all scenarios, 24 countries are expected to initiate and complete at least one pOCV by 2035. Assuming a 4-phase pOCV with slow country adoption, 17 countries initiate second round preventive vaccination compared to 24 countries in scenarios with fast adoption. For both slow and fast adoption, 50% less countries are able to initiate third round preventive vaccination compared to a 3-phase pOCV. Stretching pOCV over more years also means supply increases faster than demand, while the annual and overall market size is smaller. As a result, suppliers may leave the market if attractiveness falls below a threshold. In scenario 6, these dynamics are expected to renew delays in fulfilling OCV demand in preventive settings starting in 2033. Regardless of the number of phases in a pOCV, more time is needed to close the supply gap when country adoption is accelerated.

## 4.4. Impact of market shaping strategies

Although scenario 4 is defined as the 'preferred' demand scenario to accelerate cholera control efforts, the behavior of OCV supply and demand depend on multiple dynamics not previously accounted for. These have an impact on the volume and timing of OCV requests, influencing market dynamics and thus the ability to deliver sufficient doses in a timely manner.

**4.4.1. Feedback dynamics under budget constraints.** The preferred demand scenario, with accelerated market entry of suppliers and ambitious multisectoral interventions, has a programmatic cost of ~USD 3.3 billion. As seen in scenario 10, reducing available funds through 2034 by ~15% to USD 2.75 billion could translate to a 20% gap in doses delivered to preventive settings (Table 5). Since emergency requests for OCV are prioritized and budget constraints are only felt late (~2032) in the simulation when outbreaks have significantly declined, a 0.5% gap is expected in doses delivered to reactive settings. However, this can be concerning if new transmission chains spread due to the inability to rapidly deploy OCV during outbreaks. Several interventions could help remedy funding gaps to meet country needs around pOCV. Under budget constraints, RDT and PCP are effective strategies to reduce programmatic gaps. By targeting the most at-risk populations within PAMIs, deployment of RDT (scenario 11) leads to a 16% decrease in

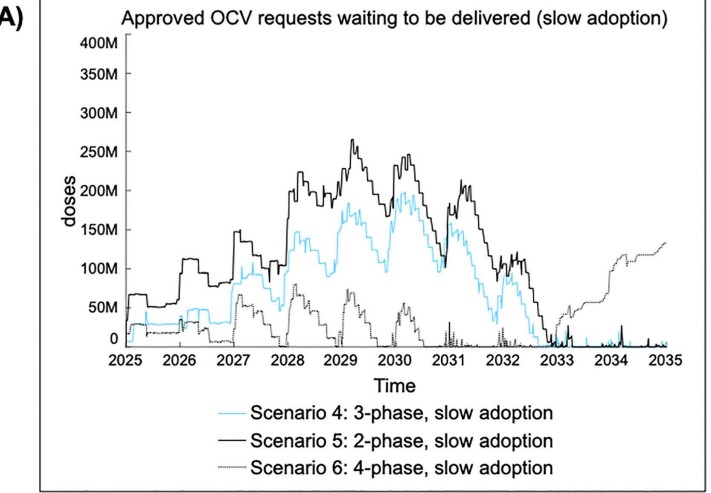
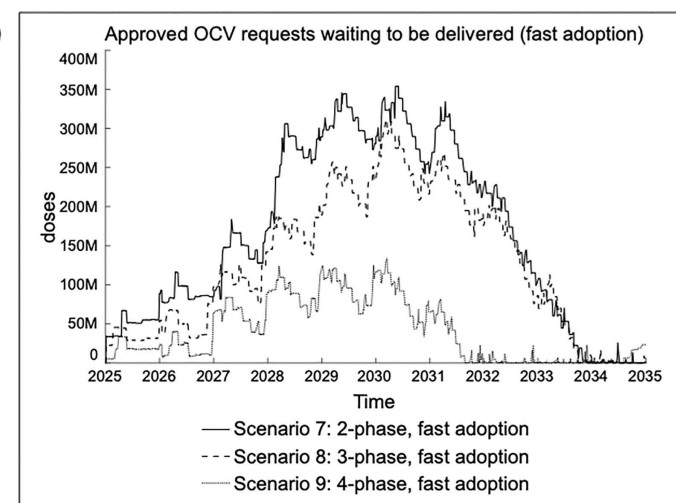
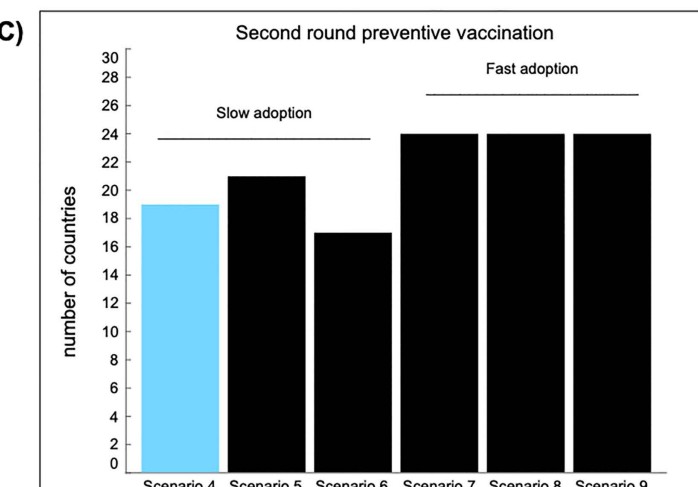
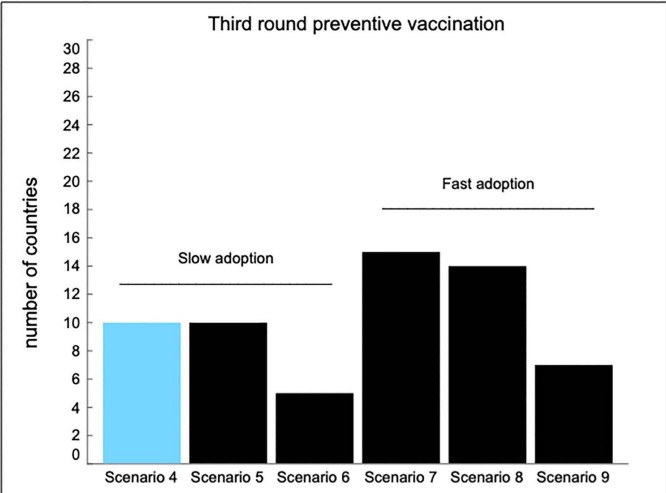

**Fig 4. Key performance indicators for scenarios 5-9, relative to scenario 4 (blue).** Approved OCV requests waiting to be delivered to countries are shown under slow (A) and fast (B) adoption of prevention vaccination in endemic countries. Also, differences can be seen in the number of countries initiating a second (C) and third (D) multi-year phased preventive vaccination program.

preventive demand and 47% reduction in unfilled DLs relative to scenario 10. In scenario 12, accounting for PCP leads to an even greater reduction in unfulfilled DLs, while delivering more doses overall (Fig F in S1 Appendix). In scenario 13, the effect of OB due to the accumulation of unfulfilled DLs leads to a smaller change than the individual effect of RDT or PCP. However, this is highly sensitive to the threshold for unfulfilled DLs that drives behavior, assumed to be 150 million doses. Combining all three interventions in scenario 14 provides the greatest alignment between supply and demand across the simulation period (Fig G in S1 Appendix).

**4.4.2. Policies for resuming full OCV use.** Scenarios 14–20 consider the relative impact of different inventory policies for resuming pOCV and BOOST, given previously-defined budget constraints and the combined effect of RDT, PCP, and OB. All inventory policy combinations tested for both decisions are comparable when looking at the ability to fulfill demand in the long-term. However, tradeoffs emerge when considering both pOCV and BOOST together. When keeping the pOCV policy constant, increasing the threshold for resuming BOOST leads to greater reliance on a 1-dose

**Table 5. Impact of different interventions to overcome budget constraints (BC) – including rapid diagnostic testing (RDT), price competition of procurement (PCP), and ordering behavior of countries (OB) – as well as effect of policies for resuming full programmatic use of OCV.**

| Scenario | Impact on OCV dynamics in reactive setting | | | Impact on OCV dynamics in preventive setting | | | Overall OCV market performance | | |
|---|---|---|---|---|---|---|---|---|---|
| | [a] $R_r$ (million) | [b] $D_r$ (%) | [c] $U_r$ (%) | [d] $R_p$ (million) | [e] $D_p$ (%) | [f] $U_p$ (%) | [g] $T_r$ (%) | [h] $T_p$ (%) | [i] $T_s$ (%) |
| 10 (BC) | 416 | 99.5 | 0.5 | 1142 | 80.0 | 20.0 | n/a | 29.9 | 75.9 |
| 11 (BC+RDT) | 438 | 99.8 | 0.2 | 985 | 89.3 | 10.7 | | 39.4 | 78.5 |
| 12 (BC+PCP) | 435 | 100 | 0 | 1153 | 92.8 | 7.2 | | 26.6 | 74.0 |
| 13 (BC+OB) | 393 | 99.7 | 0.3 | 1138 | 82.7 | 17.3 | | 30.5 | 74.7 |
| 14 (BC+RDT+PCP+OB) | 410 | 100 | 0 | 967 | 99.3 | 0.7 | | 43.1 | 80.2 |
| 15 | 573 | 100 | 0 | 822 | 99.6 | 0.4 | 76.7 | 30.5 | 76.7 |
| 16 | 451 | 100 | 0 | 987 | 99.9 | 0.1 | 53.9 | 35.7 | 75.9 |
| 17 | 557 | 100 | 0 | 914 | 98.6 | 1.4 | 64 | 31.4 | 64.0 |
| 18 | 503 | 100 | 0 | 918 | 98.9 | 1.1 | 52.5 | 36.7 | 70.7 |
| 19 | 458 | 100 | 0 | 972 | 99.8 | 0.2 | 50.1 | 78.9 | 50.1 |
| 20 | 436 | 100 | 0 | 966 | 100 | 0 | 48.4 | 81.0 | 52.8 |

*Note:* [a]$R_r$: Doses requested in reactive settings; [b]$D_r$: Percent of doses requested in reactive settings that have been delivered; [c]$U_r$: Percent of doses requested in reactive settings with unfulfilled decision letters; [d]$R_p$: Doses requested in preventive settings; [e]$D_p$: Percent of doses requested in preventive settings that have been delivered; [f]$U_p$: Percent of doses requested in preventive settings with unfulfilled decision letters; [g]$T_r$: Fraction of forecasted simulation time (2025–2035) for which 2-dose reactive vaccination is allowed; [h]$T_p$: Fraction of forecasted simulation time for which preventive vaccination is allowed; [i]$T_s$: Fraction of simulation time for which available inventory is above the stockpile target level. Color legends indicate the relative performance of interventions and policies, from worst (red) to best (green) for different key performance indicators.

strategy. Therefore, the overall demand in reactive settings decreases, as seen in scenarios 16, 18, and 20. A pOCV policy based on total WIP, when the expected inventory within ~10 weeks is 60 million doses (12 time the S*), allows the supply gap to be closed the fastest (**Fig H in** S1 Appendix). Compared to decisions based on available inventory in stock, policies that take the anticipated replenishment of supply into account maximize the time when fulfilling orders for pOCV is possible. However, an unintended consequence of policies based on the replenishment rate (incoming stock in transit) is the relative reduction in the ability to maintain the inventory level above S*. Under these policies, available inventory is used more quickly in anticipation of rapid replenishment. Although better for accelerating preventive efforts, it may lead to some delays in outbreak response if emergency requests are made just after large OCV deployments as part of planned campaigns.

### 4.5. Bridging research and practice with an interactive tool

Development of a user interface had several benefits. The visual-based nature of the ILE (Fig 5) allowed validation of uncertainty in parameters and ranges. It also encouraged discussions on realistic and likely future scenarios, serving as a sounding board for interventions users might want to test. This led to refinement of both the model and interface. From a modeler's perspective, it also helped to accelerate the validation of simulated behavior, including testing extreme conditions. Stakeholders were enthusiastic about the ILE and the ability to explore a broad range of scenarios, with a high degree of flexibility in defining parameters. Given the ILE's potential to facilitate dialogue among stakeholders involved in market shaping activities, they recommended further pursuing its development.

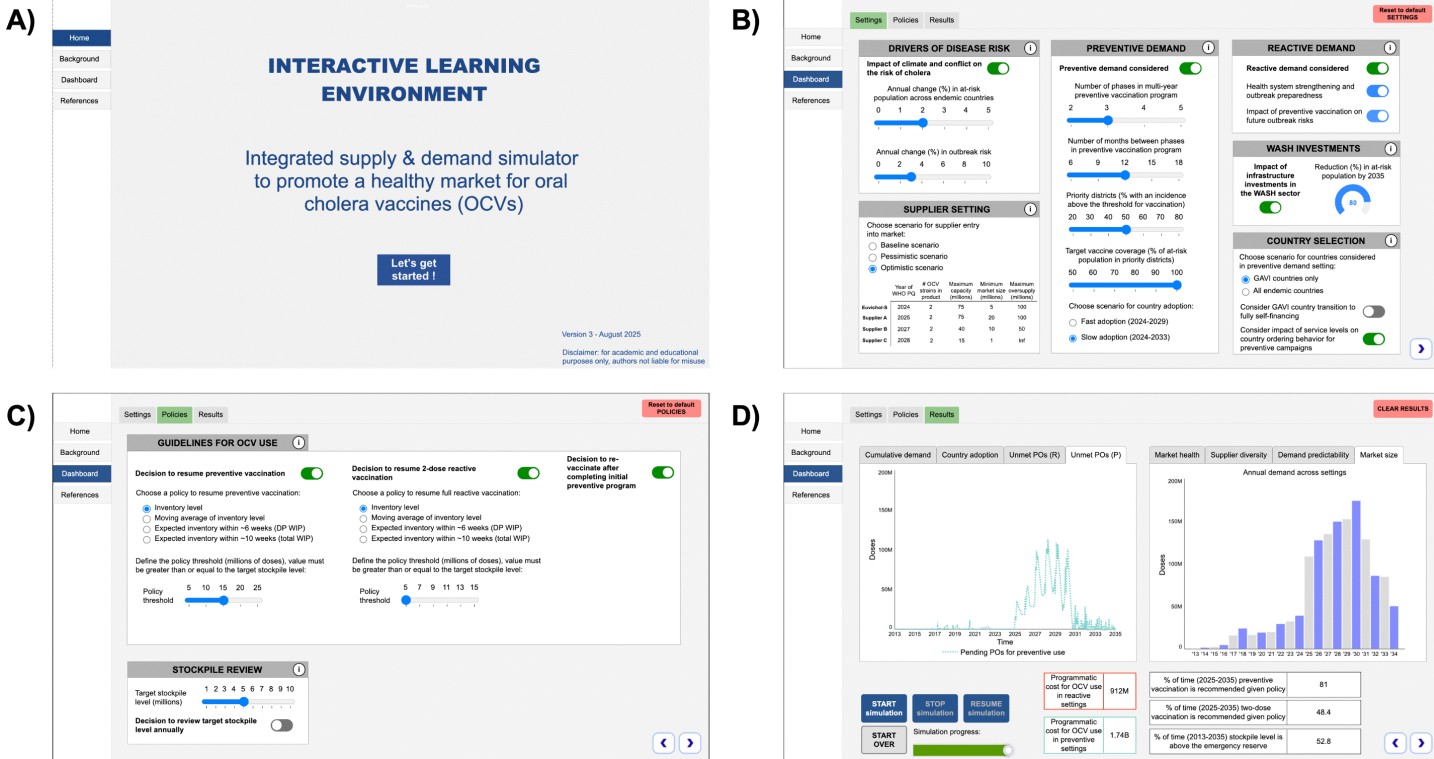

**Fig 5. Illustrative screenshots of the interactive user interface developed to facilitate learning through model simulation.** A) Landing page and menu options; B) Supply and demand settings; C) Interventions and policies; D) Dashboard of key performance indicators related to OCV market dynamics.

## 5. Discussion

This study provides insights into complex, time-dependent dynamics between OCV supply and demand. It adds value to existing literature by presenting a model that reflects the scope of decisions, processes, and stakeholders relevant to OCV market dynamics. The model improves on existing work by considering, at a strategic level, interdependencies between globally aggregate reactive demand, preventive demand across individual endemic countries, and evolving supply. It presents a novel methodology for integrating demand forecasting, production and supply modeling, and behavioral feedback (e.g., country ordering behavior, market attractiveness, budget constraints). Since the model reflects stakeholder perspectives and priorities across the OCV market, it provides a common framework to encourage dialogue. Together with the user interface, the model serves as learning tool for identifying challenges, designing market shaping strategies, and testing interventions. Although vaccine market dynamics are highly uncertain and complex, this work helps consolidate drivers of behavior with the aim of promoting greater transparency and coordination for long-term market health.

The dynamics presented have direct managerial and policy implications on the implementation of OCV-related strategies, a key pillar of GTFCC's strategy to accelerate global cholera control. Specifically, they highlight the importance of comprehensive, ambitious, and sustained prevention measures. Adopting preventive vaccination, with complementary multisectoral interventions, helps prevent outbreaks and drives down OCV demand in reactive settings. Additionally, sustaining efforts through re-vaccination of at-risk populations in endemic countries allows for greater predictability of demand, market attractiveness, and the ability for suppliers to plan production. However, given current supply constraints

and assumptions around country adoption of preventive vaccination, continued scale-up of production capacity is needed through 2030 to meet demand. Programmatic design, such as prolonging multi-phase preventive vaccination programs, can help smooth demand but may have implications on market attractiveness. Furthermore, budget constraints risk derailing vaccination efforts, leading to unfulfilled orders and less overall coverage. Market shaping strategies, such as price competition of procurement and innovative policies to resume full OCV use, are able to ensure alignment of OCV supply with demand, as well as fully meet programmatic ambitions of countries.

Although the analysis showed that large investments in OCV and WASH are needed over the next decade, they are expected to generate substantially larger long-term value. The ability to control cholera outbreaks and improve large-scale infrastructure will lead to benefits in health, including reducing the burden of other water-borne diseases, as well as related sectors of society and the economy [51,52]. Multisectoral actions targeting the most vulnerable are critical to overcome deeply-rooted inequalities. For OCV, ensuring sustained funding through GAVI replenishment is critical to safeguard global immunization efforts to meet target vaccine coverage levels. For multisectoral interventions, although aid is important to bridge funding gaps, greater domestic resource mobilization is needed to increase investments in strengthening health systems. The level of investments should align with risk-based, evidence-informed priorities defined in NCPs [53].

This study had several limitations, pointing to opportunities for future research. First, some unintended consequences of decisions may not be captured. For example, lower procurement price helps overcome budget constraints but may increase reluctance of existing suppliers to stay in the market and new ones from entering. Second, inherent evidence gaps may lead model behavior to deviate from real-world dynamics. For example, underreporting of cholera in endemic countries could lead to an underestimation of demand and overestimation of the impact of OCV. Furthermore, more evidence is needed on the impact of dose intervals on vaccine effectiveness [54], especially in endemic countries where frequent re-exposure to cholera may lead to natural immunity or serve as a natural booster [55]. There is also little evidence on the relationship between preventive vaccination and outbreak risks, especially considering potential long-range, cross-border transmission. These and other knowledge gaps are defined in a prioritized research agenda for OCV published in 2021 [56]. Although the model makes use of heuristics validated by stakeholders, future work could consider epidemiological dynamics of cholera transmission and flexible dosing schedules. Such an extension of the model could also capture the effect of novel *V. cholerae* variants with different transmissibility and virulence. Third, better understanding country-specific drivers of adopting preventive measures would be valuable. Future work could include modeling decisions around prioritizing national cholera control efforts, including changing national income levels, GAVI co-financing, and the capacity for conducting large-scale OCV campaigns.

From the supply side, more complex incentives, risk thresholds, and internal decision-making processes might drive perceived market attractiveness and supplier decisions to enter or leave the market. Although static supply settings are used, innovative financing mechanism and procurement processes could incentivize more rapid and larger investments from suppliers in OCV development and production [57]. Additionally, OCV production is defined annually based on the maximum available capacity for each supplier. However, future work could explore the impact of dynamic production planning on the ability to close supply gaps while reducing excess inventory. Additionally, a broader set of supplier settings could be considered, including entry of new OCV products with less or more strains, as reported in literature. This would impact production throughput and availability of supply. Extensions of the model could explore strategic, seasonal pre-positioning of stocks in countries where outbreaks are likely to occur, rather than keeping them in the global OCV stockpile where doses may be used in preventive settings. In the next decade, cholera conjugate vaccines are expected to enter the market, providing longer duration of vaccine-induced immunity, especially for children under five years of age. Future work could explore the impact of these next-generation vaccines on OCV demand and long-term cholera control efforts, especially if countries consider integrating them in routine vaccination schedules.

Further work on the interactive user interface is also important for continuous learning and improvement. For instance, the single-player simulation and user guide could be turned into a multi-player game, where different users only define settings which they can control, for example, supply or demand, depending on expected stakeholder roles and capacities. This could highlight the importance of information sharing, transparency, and collaborative design of interventions to improve market health. Finally, the methodological approach could be applied to other vaccine markets, especially those against epidemic-prone pathogens with complex dynamics driven by uncertain demand and evolving global policies around their programmatic use.

## Supporting information

**S1 Appendix. Table A: Interviewees categorized by organization type. Table B**: Defining key KPIs based on stakeholder discussions. **Table C**: Tradeoffs across key KPIs for scenarios 4–7. **Table D**: Overview of model parameters – regulatory approval subsystem. **Table E**: Values for model parameters arrayed by product – regulatory approval subsystem (optimistic & pessimistic). **Table F**: Overview of model parameters – global OCV policy subsystem. **Table G**: Overview of model parameters – demand subsystem. **Table H**: Impact of health system strengthening and outbreak preparedness. **Table I**: Seasonal distribution of reactive orders per month. **Table J**: Country specific data. **Table K**: Overview of model parameters – product development subsystem. **Table L**: Values for model parameters arrayed by product. **Table M:** Overview of model parameters – production and supply subsystem. **Table N:** Overview of model parameters – order fulfilment subsystem. **Table O**: Overview of model parameters – climate and conflict subsystem. **Fig A**: Evolution of WHO recommendations and OCV supplier landscape, with a focus on key milestones around technology transfer and WHO pre-qualification. The mapping also includes publicly reported OCV development efforts. OCVs have only been recommended as part of broader cholera control strategies since 2010. **Fig B**: Comparison of daily simulated OCV inventory levels (grey) to weekly reported OCV inventory levels (green) reported by UNICEF (2021–2024) [58]. The target level for the emergency OCV stockpile (red) is 5 million doses during the period presented. **Fig C**: Ratio of annual doses delivered to reactive (red) versus preventive (blue) settings for scenario 1–4. **Fig D**: Annual doses requested across all demand settings, scenarios 1–4. **Fig E**: A) Total programmatic cost, including OCV procurement and operational support for campaigns, for the entire simulation (2013–2035) for scenarios 1–4. B) Costs during the GAVI 5.0 funding period (2021–2025). C) Costs during the GAVI 6.0 funding period (2026–2030). D) Costs during the GAVI 7.0 funding period (2031–2035). **Fig F**: Cumulative doses requested for scenarios 10–14, compared to scenario 4 (no budget constraint), across reactive (A), preventive (B), and all (C) settings. **Fig G**: Approved requests that have not been delivered for scenarios 10–14 compared to scenario 4 (no budget constraint). **Fig H**: Approved request that have not been delivered for scenarios 14–20, considering different polices for resuming preventive vaccination and two-dose reactive vaccination. **Fig I**: Seasonality of historic ICG requests, reported monthly and for each year between 2013–2024 (left axis, colored points) and the average proportion of requests over all years happening in each month (right axis, red line). **Fig J**: Geographic distribution of historic ICG requests reported by country, cumulative between 2013–2024. **Fig K**: Model structure of key stocks and flows – reactive demand. **Fig L**: Model structure of key stocks and flows – preventive demand. **Fig M**: Model structure of key stocks and flows – production and supply.
(DOCX)

## Acknowledgments

The authors would like to express their gratitude to all the interviewees who generously contributed their time to share experiences and perspectives on the research.

## Author contributions

**Conceptualization:** Donovan Guttieres, Carla Van Riet, Nico Vandaele, Catherine Decouttere.

**Data curation:** Donovan Guttieres.

**Formal analysis:** Donovan Guttieres.

**Funding acquisition:** Nico Vandaele, Catherine Decouttere.

**Methodology:** Donovan Guttieres.

**Project administration:** Donovan Guttieres.

**Software:** Donovan Guttieres.

**Supervision:** Carla Van Riet.

**Validation:** Donovan Guttieres, Carla Van Riet, Nico Vandaele, Catherine Decouttere.

**Writing – original draft:** Donovan Guttieres.

**Writing – review & editing:** Donovan Guttieres, Carla Van Riet, Nico Vandaele, Catherine Decouttere.

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
