## [Decision Letter · Decision Letter 0]

16 Dec 2025

Dear Dr. Guttieres,

Thank you for submitting your manuscript to PLOS Neglected Tropical Diseases. After careful consideration, we feel that it has merit but does not fully meet PLOS Neglected Tropical Diseases's publication criteria as it currently stands. Therefore, we invite you to submit a revised version of the manuscript that addresses the points raised during the review process.

* A letter that responds to each point raised by the editor and reviewer(s). You should upload this letter as a separate file labeled 'Response to Reviewers '. This file does not need to include responses to any formatting updates and technical items listed in the 'Journal Requirements' section below.

* A marked-up copy of your manuscript that highlights changes made to the original version. You should upload this as a separate file labeled 'Revised Manuscript with Track Changes '.

* An unmarked version of your revised paper without tracked changes. You should upload this as a separate file labeled 'Manuscript '.

We look forward to receiving your revised manuscript.

Kind regards,

Jeffrey H Withey

Academic Editor

Ana LTO Nascimento

Section Editor

Shaden Kamhawi

co-Editor-in-Chief

Paul Brindley

co-Editor-in-Chief

**Additional Editor Comments:**

The reviewers were quite positive about your manuscript and it should be acceptable with minor modifications. Please respond to the comments of the reviewers and indicate where you have made changes in the revised manuscript.

**Journal Requirements:**

1) Please provide an Author Summary. This should appear in your manuscript between the Abstract (if applicable) and the Introduction, and should be 150-200 words long. The aim should be to make your findings accessible to a wide audience that includes both scientists and non-scientists. Sample summaries can be found on our website under Submission Guidelines:

2) Your manuscript is missing the following sections: Methods. Please ensure all required sections are present and in the correct order. Make sure section heading levels are clearly indicated in the manuscript text, and limit sub-sections to 3 heading levels. An outline of the required sections can be consulted in our submission guidelines here:

Potential Copyright Issues:

i) Please confirm (a) that you are the photographer of 5A, or (b) provide written permission from the photographer to publish the photo(s) under our CC BY 4.0 license.

6) Kindly revise your competing statement in the online submission form to align with the journal's style guidelines: 'The authors declare that there are no competing interests.'

**Reviewers' comments:**

Reviewer's Responses to Questions

**Key Review Criteria Required for Acceptance?**

**Methods**

-Are the objectives of the study clearly articulated with a clear testable hypothesis stated?

-Is the study design appropriate to address the stated objectives?

-Is the population clearly described and appropriate for the hypothesis being tested?

-Is the sample size sufficient to ensure adequate power to address the hypothesis being tested?

-Were correct statistical analysis used to support conclusions?

-Are there concerns about ethical or regulatory requirements being met?

Reviewer #1: The suitability of SD modeling is explained, but the specific reason for using SD over other modeling approaches is not justified.

Please provide a comparative justification or approach that demonstrates its ability to capture feedback relationships and system-level policy effects.

Reviewer #2: Yes, but complex model description

**Results**

-Does the analysis presented match the analysis plan?

-Are the results clearly and completely presented?

-Are the figures (Tables, Images) of sufficient quality for clarity?

Reviewer #1: (No Response)

Reviewer #2: Yes, but no easily described results

**Conclusions**

-Are the conclusions supported by the data presented?

-Are the limitations of analysis clearly described?

-Do the authors discuss how these data can be helpful to advance our understanding of the topic under study?

-Is public health relevance addressed?

Reviewer #1: (No Response)

Reviewer #2: Yes, good

**Editorial and Data Presentation Modifications?**

Reviewer #1: (No Response)

Reviewer #2: A long manuscript with many abbreviations, scenarios and assumption. But I would say i have just minor comments. Perhaps some fact-boxes - with scenario assumptions would make the manuscript more easily read and understood.

**Summary and General Comments**

Reviewer #1: (No Response)

Reviewer #2: Impressive work which include stakeholders interviews and validation make this model feel solid.

The manuscript contains an elaborate and ambitious complex model design, including stakeholders input and validation and would be a welcome addition to strategy planners and manufacturers of OPV. I am not a modeller, but consider the approaches reasonable – though complex with multiple scenarios.

I find the text well written and discussion insightful.

I have just a few comments.

First, this model considers the role of OCV, as well as complementary multisectoral interventions (as stated in the introduction), and your efforts and model is impressive. I understand that this comment may fall outside of your main scope, but I still miss an assumption of how many cases and/or deaths could be expected with the different scenarios – if that is possible. You call this later in the text (page19, line 552) significant reduction requested in reactive settings – and this could perhaps be translated to assumed number of prevented cases. If an increased cost is to be swallowed it would perhaps be important to show saved lifes and cases – even if that is not an easy task, and pit out how much closed to the goals set in the Global Task Force on Cholera Control one can come. You have commented something concering this issue in the discussion but it could perhaps be more highlighted in the abstract.

I am not totally confident that “phases” (tabl 3) are sufficiently explained for an uninformed reader.

It is good that assumptions into the model are referenced, but I can´t find why 3 further OPV will be available at 2028. Is it just an assumption or based on developer’s prognosis?

A small misspelling was discovered at page 12, line 369 concerning the timespan of the simulation period.

At page 19 line 546 it is stated that 881 m doses are requested, but later that 75 m doses is sufficient. I guess that the first number refers to the whole simulation period. If that is correct please make clear.

PLOS authors have the option to publish the peer review history of their article (what does this mean? ). If published, this will include your full peer review and any attached files.

**Do you want your identity to be public for this peer review?** For information about this choice, including consent withdrawal, please see our Privacy Policy .

Reviewer #1: **Yes:** Manfred Dakorah Asiedu

Reviewer #2: No

**Figure resubmission:**
---

## [Editor Report · Decision Letter 1]

2 Feb 2026

Dear Dr. Guttieres,

We are pleased to inform you that your manuscript 'Strategies for achieving a healthy oral cholera vaccine market: Model-enabled scenario exploration of supply and demand dynamics' has been provisionally accepted for publication in PLOS Neglected Tropical Diseases.

Best regards,

Jeffrey H. Withey

Academic Editor

Ana LTO Nascimento

Section Editor

Shaden Kamhawi

co-Editor-in-Chief

Paul Brindley

co-Editor-in-Chief

p.p1 {margin: 0.0px 0.0px 0.0px 0.0px; line-height: 16.0px; font: 14.0px Arial; color: #323333; -webkit-text-stroke: #323333}span.s1 {font-kerning: non

---

## [Editor Report · Acceptance letter]

Dear Dr. Guttieres,

We are delighted to inform you that your manuscript, "Strategies for achieving a healthy oral cholera vaccine market: Model-enabled scenario exploration of supply and demand dynamics," has been formally accepted for publication in PLOS Neglected Tropical Diseases.

Best regards,

Shaden Kamhawi

co-Editor-in-Chief

Paul Brindley

co-Editor-in-Chief
